# Changes in Essential Fatty Acids and Ileal Genes Associated with Metabolizing Enzymes and Fatty Acid Transporters in Rodent Models of Cystic Fibrosis

**DOI:** 10.3390/ijms24087194

**Published:** 2023-04-13

**Authors:** Nirajan Shrestha, Nathan Rout-Pitt, Alexandra McCarron, Courtney A. Jackson, Andrew C. Bulmer, Andrew J. McAinch, Martin Donnelley, David W. Parsons, Deanne H. Hryciw

**Affiliations:** 1School of Pharmacy and Medical Sciences, Griffith University, Southport, QLD 4215, Australia; 2Robinson Research Institute, University of Adelaide, Adelaide, SA 5005, Australia; 3Adelaide Medical School, University of Adelaide, Adelaide, SA 5001, Australia; 4Respiratory and Sleep Medicine, Women’s and Children’s Hospital, 72 King William Road, North Adelaide, SA 5006, Australia; 5School of Environment and Science, Griffith University, Nathan, QLD 4215, Australia; 6Institute for Health and Sport, Victoria University, Melbourne, VIC 3000, Australia; 7Australian Institute for Musculoskeletal Science (AIMSS), Victoria University, St. Albans, VIC 3021, Australia; 8Griffith Institute for Drug Discovery, Griffith University, Nathan, QLD 4111, Australia

**Keywords:** cystic fibrosis, fatty acids, intestine

## Abstract

Cystic fibrosis (CF), the result of mutations in the CF transmembrane conductance regulator (CFTR), causes essential fatty acid deficiency. The aim of this study was to characterize fatty acid handling in two rodent models of CF; one strain which harbors the loss of phenylalanine at position 508 (Phe508del) in CFTR and the other lacks functional CFTR (510X). Fatty acid concentrations were determined using gas chromatography in serum from Phe508del and 510X rats. The relative expression of genes responsible for fatty acid transport and metabolism were quantified using real-time PCR. Ileal tissue morphology was assessed histologically. There was an age-dependent decrease in eicosapentaenoic acid and the linoleic acid:α-linolenic acid ratio, a genotype-dependent decrease in docosapentaenoic acid (n-3) and an increase in the arachidonic acid:docosahexaenoic acid ratio in Phe508del rat serum, which was not observed in 510X rats. In the ileum, *Cftr* mRNA was increased in Phe508del rats but decreased in 510X rats. Further, *Elvol2*, *Slc27a1*, *Slc27a2* and *Got2* mRNA were increased in Phe508del rats only. As assessed by Sirius Red staining, collagen was increased in Phe508del and 510X ileum. Thus, CF rat models exhibit alterations in the concentration of circulating fatty acids, which may be due to altered transport and metabolism, in addition to fibrosis and microscopic structural changes in the ileum.

## 1. Introduction

Cystic fibrosis (CF) is an autosomal recessive disorder caused by mutations in the gene encoding the CF transmembrane conductance regulator (*Cftr*) [1]. Mutations in *Cftr* are categorised into six different classes based on their effects on CFTR function [2]. The most common mutation associated with CF is the Phe508del mutation in *Cftr,* which is a Class II variant that results in the loss of the phenylalanine at position 508 of CFTR, for which 70% of patients are homozygous. A significant proportion of the Phe508del CFTR protein is retained in the endoplasmic reticulum and fails to traffic to the cell membrane [1]. 

CFTR dysfunction affects multiple epithelial organs including the gut, pancreas and reproductive tract, with the most significant pathology associated with the lungs [3]. Notably, CF is also characterized by an increased inflammatory status [4]. In addition, CF is associated with abnormal polyunsaturated fatty acid (PUFA) metabolism, which ultimately results in an essential fatty acid (EFA) deficiency [5], independent of pancreatic insufficiency [6]. The nutritional status of patients with CF impacts morbidity and mortality [7]. Specifically, EFA status in patients with CF is closely linked to the risk of pulmonary infection, the most significant pathology associated with CF [7]. The effects of EFA imbalance on other tissues in CF is not well understood.

Within the diet, EFAs are present in the form of triglycerides. EFAs are absorbed and then transported in the blood. EFAs can be assimilated in various organs including brain, retina, skeletal muscles, adipose and other tissues [8]. EFAs can undergo (a) β-oxidation to provide energy in the form of adenosine triphosphate (ATP), (b) esterification into cellular lipids including triglycerides, cholesterol esters and phospholipids, and (c) metabolism into longer-chain fatty acids by desaturation and elongation reactions [8]. Uptake of EFAs in tissues occurs via specific transmembrane (fatty acid transport (FATP)) and intracellular (fatty acid binding (FABP)) proteins. In the intestine, Slc27a1, Slc27a2 and Slc27a4 are responsible for FA transport across the membrane, and Got2 and Cd36 are responsible for intracellular FA transport [9]. Once absorbed by the intestine and liver, EFAs can be metabolized and stored or transported back into circulation [9]. Metabolism of EFAs is abundant in the liver and, to a lesser extent, in other tissues [8].

PUFAs can be classified into two main types—omega-3 (n-3) and omega-6 (n-6) FA, which contain the EFAs α-linolenic acid (ALA) and linoleic acid (LA), respectively [10]. ALA and LA cannot be synthesized in humans and must be obtained via the diet. LA can be metabolized into γ-linolenic acid (GLA) and arachidonic acid (AA), while ALA is metabolized into docosahexaenoic acids (DHA) and eicosapentaenoic acid (EPA). Metabolism of LA and ALA occurs via the same desaturation and elongation enzymes; namely, fatty acid desaturase (FADS) FADS1, FADS2 (also referred to as Δ5 desaturase and Δ6 desaturase, respectively) and elongation of very long-chain fatty acids protein (ELOVL) ELOVL2 and ELOVL5 [11]. Patients with CF routinely have deficiencies in EFAs [12]—specifically, decreased circulating concentrations of LA and increased AA:DHA [13]. The identification of a reduction in circulating LA in patients with CF is long established [14].

Limited research has investigated the molecular mechanism that gives rise to EFA deficiency [10] in animal models of CF [15]. In our recent review, we summarized the EFA status in animal models of CF [10]. Within these models, where the EFA status has been characterized, the EFA concentrations do not mimic what is observed in patients with CF [10]. A recently developed animal model, the CF rat, has several characteristics similar to patients with CF. The Phe508del mutation results in a misfolded CFTR protein, with minimal CFTR protein that is trafficked to the cell membrane. Any Phe508del CFTR protein has defective gating and a short half-life. At 1 month of age, CF rats with the Phe508del mutation and the 510X mutation have impaired survival, reduced body weight, intestinal obstruction, and histopathologic changes in the trachea, large intestine and pancreas [16]. Further, in the bronchioles, the CFTR protein in the Phe508del and 510X rats is localized in the cytoplasm, with reduced CFTR protein expression in the 510X rats [16]. However, the EFA status in the CF Phe508del and 510X rats is currently unknown. The hypothesis of this study is that CF Phe508del and 510X rats demonstrate changes in EFA concentrations due to altered expression in genes responsible for EFA transport and metabolism, and histological changes to the ileum, similar to patients with CF. 

## 2. Results

### 2.1. Fatty Acid Concentrations in the Serum of CF Rat Models

There was an age-dependent decrease in eicosapentaenoic acid (EPA) and the linoleic acid:α-linolenic acid ratio (Table 1, *p* < 0.05, *n* = 3–4). There was also a genotype-dependent decrease in docosapentaenoic acid (DPA:n-3) for Phe508del rats and an increase in the arachidonic acid:docosahexaenoic acid (AA:DHA) ratio in Phe508del rat serum (Table 1, *p* < 0.05, *n* = 3–4). Other fatty acid concentrations were unaltered (Table 1, *n* = 3–4). 

### 2.2. Gene Expression in the Ileum of CF Rat Models

*Cftr* expression in the ileum was significantly elevated in the 1-month-old Phe508del rats (*p* < 0.05, *n* = 6), while expression was significantly decreased in the 510X rats compared to the wild type (*p* < 0.05, *n* = 6) (Figure 1). In the ileum, *Elvol2*, *Scl27a1*, *Scl27a2* and *GOT2* were significantly increased in Phe508del rats compared to the wild type (Figure 2, *p* < 0.05, *n* = 5–6), but expression in 510X rats was not different. *Elovl5*, *Fads1*, *Fads2*, *Slc27a4*, *Faah*, and *CD36* gene expression were not significantly different between the groups (Figure 2).

### 2.3. Protein Filament Gene Expression and Histological Analysis of Ileum Tissue of CF Rat Models

In 2-month-old Phe508del and 510X rats, abundant mucus in markedly dilated crypts was observed, with mucus extrusion into the lumen (Figure 3), similar to our previous study on 1-month-old rats [16]. The ileal mean intensity in PAS staining was not different between the groups. The height of the villi was significantly increased in 510X (370.79 ± 82.77 nm) rats compared with Phe508del (165.84 ± 47.20 nm) and the wild type (236.64 ± 53.54 nm; *p* < 0.05, *n* = 6). Mean intensity in Sirius Red staining was increased in Phe508del (200.19 ± 2.28) and 510X (201.23 ± 0.42) rats compared with the wild type (179.95 ± 9.71) (*p* < 0.05, *n* = 6). Collagen type 1a1 (Col1a1) and collagen type 3 a 1 (Col3a1) are fibrotic genes that are thought to contribute to fibrosis development in pancreas, lungs and intestine in patients with CF [17]. When we assessed the gene expression of *Col1a1* and *Col3a1*; however, there was no significant difference between the groups (Figure 2). 

## 3. Discussion

This is the first study to characterize the EFA status in CF adolescent and young adult rats. Similar to CF patients [13], there was an increase in the AA:DHA in Phe508del rat serum, which interestingly was not observed in 510X rats. There was also an age-dependent decrease in eicosapentaenoic acid and the LA:ALA ratio, genotype-dependent decrease in DPA (n-3). This may in part be due to the increase in Elvol2 in the ileum of Phe508del rats. Previous research using a global knockout Elovl2 mouse demonstrated a reduction in serum and liver DHA [18]. Therefore, tissue concentrations of Elovl2 may contribute to the serum concentrations of DHA and thus the AA:DHA ratio. Interestingly, Phe508del rats also demonstrated an increase in *Slc27a1, Slc27a2* and *Got2*, which transport fatty acids. Research on *Slc27a1, Slc27a2* and *Got2* in the ileum is limited; however, *Slc27a1* in rat ileum is reduced in rats fed a high fat diet [19].

Gastrointestinal changes in collagen have been observed in a porcine model of CF [20]. Further, our study showed an increase in ileal collagen (as indicated by Sirius Red staining) in Phe508del and 510X ileum, suggestive of structural damage, although there were no associated changes in the expression of collagen genes *Col1a1* or *Col3a1*. In the 510X rats, the increase in villi height and mucus accumulation in the intestinal lumen is also indicative of the pathology observed in CF patients. 

Patients with CF typically present with a reduction in serum concentration of LA [14]. However, in this study we did not observe a reduction in serum concentrations of LA in CF rats. In CF mice, some models also fail to demonstrate a reduction in LA in the serum [21], while other CF mouse models do exhibit a reduction [14,22]. A recent manuscript identified a reduction in LA in CF pigs and CF ferrets [23]. This suggests a species-specific and age-specific changes in LA in animal models of CF. The requirement of LA in the diet is ~1.5% [24], with rats in this study consuming ~3% [16]. It has been shown in vitro that elevated LA in the culture media increases lung epithelial cell LA concentrations [23,25]; therefore, elevated LA in the diet may increase the concentrations in serum. Notably, the rats in this study were non-fasted, with samples collected in the light cycle, where rodents graze less. Data comparisons were made compared to wild-type rats with serum collected under the same conditions. Despite this, the fatty acid analyses may have been influenced by food consumption prior to blood sample collection. Compared to the CF mice models, with differences in EFA status based on the genetic background [10], the increased AA:DHA ratio in serum of the CF Phe508del rat suggests that this model is closer to patients with CF, despite LA concentrations in the serum being unchanged.

*Cftr* mRNA levels in the ileum of the Phe508del rat were significantly increased when compared to the wild type, while *Cftr* was significantly reduced in the 510X model. This decrease in the 510X model is consistent with our previous findings from the lung tissue, and indicates degradation of the *Cftr* transcripts via nonsense-mediated mRNA decay [16]. Although we previously found decreased mRNA levels in Phe508del rat lungs [16], increases in Phe508del *Cftr* expression have been previously observed [26], indicating that there could be tissue-specific differences. The difference in ileum *Cftr* mRNA levels suggests that despite both rat models having a CF phenotype, the defective Cftr results in different underlying molecular changes. For example, there is emerging research that suggests Phe508del tissues exhibit impaired cellular proteostasis and autophagy, due to the retention of the misfolded CFTR in the endoplasmic reticulum [27]. Thus, it is plausible that the retention of CFTR in the endoplasmic reticulum of the Phe508del rat enhances the expression of the *Cftr* mRNA in the ileum. However, examination of protein expression is necessary to explore this proposed phenomenon. Based on differing mechanisms of *Cftr* mRNA and protein handling in the Phe508del and 510X rats, differences in the CF phenotypes of the two rodent models are observed. 

Previous research has investigated the underlying molecular metabolisms that give rise to EFA deficiencies in CF. In a bronchial epithelial cell culture model of CF, there was an increase in *Fads1* and *Fads2* expression [28], which was driven by AMPK and an increase in calcium. This increase in *Fads1* and *Fads2* was reversed by supplementation with DHA [29]. Of note, these studies were performed on an immortalised bronchial epithelial cell line, which cannot maintain all the characteristics of intact in vivo CF lungs. In our study, we demonstrated an increase in *Elovl2* in the ileum of Phe508del rats. Elovl2 is required for the metabolism of EPA to DPAn-3, and AA to adrenic acid (22:4n-6) and DPAn-6. Thus, one contributor to this phenotype in the ileum of Phe508del rats could be Elovl2. Elvol2 elongates AA and EPA. As there is a decrease in DPAn-3, and an increase in the AA:DHA ratio, this may suggest that Elvolv2 has altered metabolism of DPAn-3 and DHA over AA, which could contribute to the serum concentrations of these fatty acids in this model. Future studies should look at how these specific pathways differ between wild-type, Phe508del, and knockout rat protein expression. 

One of the most widely used CF animal models is the mouse; however, CF mice lack the spontaneous lung [30] and pancreatic phenotype [31] observed in CF patients. In the context of EFA deficiency, there are conflicting data in CF mice concerning the EFA status. One study showed ileum, lung and pancreatic DHA was reduced and AA was increased in CFTR knockout mice [32] similar to the 510X rats in this study, with another study identifying no changes in tissue or plasma EFAs in Phe508del mice [33]. Compounding this, a further study showed that in Phe508del mice, LA is reduced in the pancreas, AA is increased in ileum and pancreas, and DHA is unaltered in the ileum [34]. Thus, the conflicting data concerning the Phe508del mice limit the use of this model to investigate EFA status in CF. The background strain of the CF rodent model appears to be of importance. For example, the C57B1/6J *Cftr* knockout mice had a reduction in LA, with females more severely affected than males [18]. However, *cftr*^−/−CAM^ mice did not have reduced LA in circulation (with data not separated based on sex) [33]. Further, they exhibit an increase in AA in ileum membranes [32], and a reduction in LA in the ileum [22]. 

Both rat models demonstrated colonic crypt obstruction with accumulation of mucus within months, which was similar to what was previously reported in CF rats at 1 month of age, although this was not accompanied by a quantitative increase in mucosubstances as indicated by PAS staining [16]. These intestinal histopathology changes are also observed in humans, as well as other CF animal models, including the US-generated knockout rat [35] and CF mouse models. The intestinal changes may contribute to the defective EFA transport, with further investigations warranted. This suggests that as the rats age, the intestinal phenotype becomes more compromised. An area for further research is to quantify mRNA across early adulthood in the rat models of CF. Similar to age-related changes to FA, there may be age-related changes to mRNA responsible for FA metabolism and transport. 

In conclusion, we have demonstrated, for the first time, changes in circulating EFA concentrations in the Phe508del rat, with alterations in EFA metabolic enzymes and transporters in the ileum. These changes may contribute to the overall CF phenotype observed in the rat models, and the differences in phenotype between the Phe508del and 510X rats may be in part due to differences in the expression of *Cftr* mRNA in the digestive system.

## 4. Materials and Methods

### 4.1. Animal Model

Animal studies were conducted under approval from the University of Adelaide Animal Ethics Committee (M-2017-056). From birth, all rats were maintained on a 50:50 mix of normal (6.5%) and high-fat (9%) irradiated rodent chow (Envigo 2920X and 2919, Indianapolois, IN, USA), with additional information on dietary fatty acid composition provided in the Appendix A. As described previously [15], the Phe508del model was generated using CRISPR/Cas9 and a homology-directed repair DNA template containing a TTT deletion corresponding to position 508 in the rat CFTR sequence. The 510X was developed with an 8 bp deletion in exon 11 of CFTR which results in the introduction of a premature termination codon at position 510 [16]. CF animals received 4.5% ColonLytely in their drinking water to reduce intestinal obstructions (Dendy Pharmaceuticals, Moorabbin, VIC, Australia) [15]. Tissue samples (formalin fixed, snap frozen or stored in RNAlater^®^) and serum were collected from humanely killed Phe508del and 510X CF adolescent and young adult rats at 1, 2 and 4 months of age killed during the light cycle [16], as well as wild-type aged-matched littermates. 

### 4.2. Fatty Acid Analysis

Serum fatty acid analysis was performed on non-fasted 2 and 4-month-old rats by Cardinal Bio-Research Pty Ltd. (Slacks Creek, QLD, Australia). Briefly, the rat serum was analyzed by gas chromatography (GC) with flame ionization detection. GC was carried out using a GC-2030 Gas Chromatograph (Shimadzu Corporation, Rydalmere, NSW, Australia) equipped with a SP-2560, 100 m fused silica capillary column (0.25 mm internal diameter, 0.2 μm film thickness; Restek; Leco Australia, Baulkham Hills, NSW, Australia). Fatty acids were identified by comparison with a standard mixture of fatty acids (NuCheck Prep; Adelab, Adelaide, SA, Australia). Fatty acid composition was expressed as a percent of total identified fatty acids.

### 4.3. Quantitative Real-Time Polymerase Chain Reaction (qPCR)

qPCR was performed on tissue samples collected from 1-month-old rats. Total RNA was extracted from ileum tissue stored in RNAlater^®^ using TRIzol^®^ Reagent (Thermo Fisher Scientific, Scoresby, VIC, Australia) following the manufacturer’s guidelines. The quantification and evaluation of the purity of RNA samples was assessed using the NanoDrop™ Lite spectrophotometer (Thermo Fisher Scientific, Scoresby, VIC, Australia) and RNA integrity was confirmed by visualisation of 18S and 28S ribosomal subunits upon agarose gel electrophoresis. Reverse transcription of RNA to synthesize complementary DNA was performed using the QuantiTect Reverse Transcription Kit (Qiagen, Clayton, VIC, Australia) following the manufacturer’s guidelines. Quantitative PCR (qPCR) was performed using the Fast SYBR™ Green Master Mix (Thermo Fisher Scientific, Scoresby, VIC, Australia) following the manufacturer’s guidelines, in line with the Minimum Information for Publication of Quantitative Real-Time PCR Experiments (MIQE) guidelines [36]. Primer sequences employed are included in the Appendix A. PCR samples were heated for 10 min at 50 °C followed by 3 min at 95 °C, then qPCR reactions were run for 40 cycles of 95 °C for 10 s (denaturation) and 60 °C for 60 s (combined annealing/extension) using the CFX Connect Real-Time PCR Detection System (Bio-Rad, Sydney, NSW, Australia). Gene expression was quantified using the 2^−ΔΔCq^ method normalized to the geometric mean of β-actin and cyclophilin A as reference genes and is presented as the Log base 2. The reference genes were stable across the groups. 

### 4.4. Histology

Paraffin-embedded ileal samples from 2-month-old animals were prepared [15], sectioned at 4 μm, and stained using hematoxylin and eosin (H&E), Periodic acid–Schiff (PAS) and Sirius Red. Stained sections were imaged using an Olympus BX30 microscope and quantification was performed on 6 samples from the wild type, Phe508del and 510X, using ImageJ software (Ver. 1.53k). An estimation of the intensity of glycogen and mucosubstances (PAS) and collagen (Sirius Red) was achieved using the tool measure image intensity, which quantifies the total intensity by summing all the pixel intensities.

### 4.5. Statistics

Fatty acid data were analyzed using GraphPad Prism 8.3.1 (GraphPad software, San Diego, CA, USA). Data were analyzed using two-way analysis of variance (ANOVA), with genotype and age as the 2 factors, followed by Tukey’s post hoc test. Data were analyzed using GraphPad Prism 8.3.1 (GraphPad software, San Diego, CA, USA). Following a Shapiro–Wilk test to confirm normality, data were analyzed using one-way analysis of variance (ANOVA) followed by either Tukey’s post hoc test or Dunnett’s multiple comparisons test. Data are presented as the mean ± standard error of the mean (SEM). A *p*-value < 0.05 was considered evidence of significant differences.

## Figures and Tables

**Figure 1 ijms-24-07194-f001:**
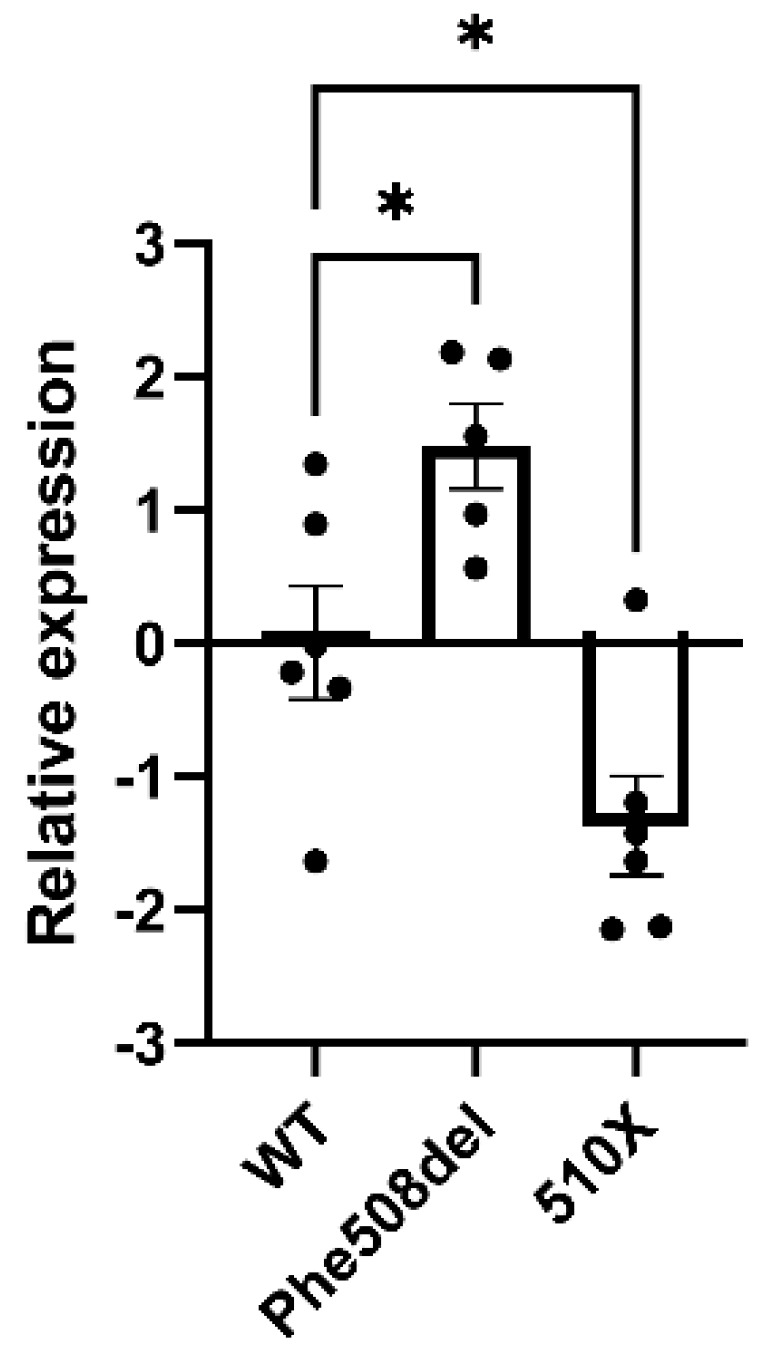
Cftr mRNA expression in ileum of 1 month old CF rats. *Cftr* expression in wild-type, Phe508del and 510X rat ileum was normalized to the average expression of the housekeeping genes β-actin and cyclophilin A. The relative expression of *Cftr* in the Phe508del and 510X ileum was compared to the wild type. Data are expressed as the mean ± SEM and analyzed by a one-way ANOVA with Dunnett’s multiple comparisons test to determine significance. * indicates *p* < 0.05, *n* = 6.

**Figure 2 ijms-24-07194-f002:**
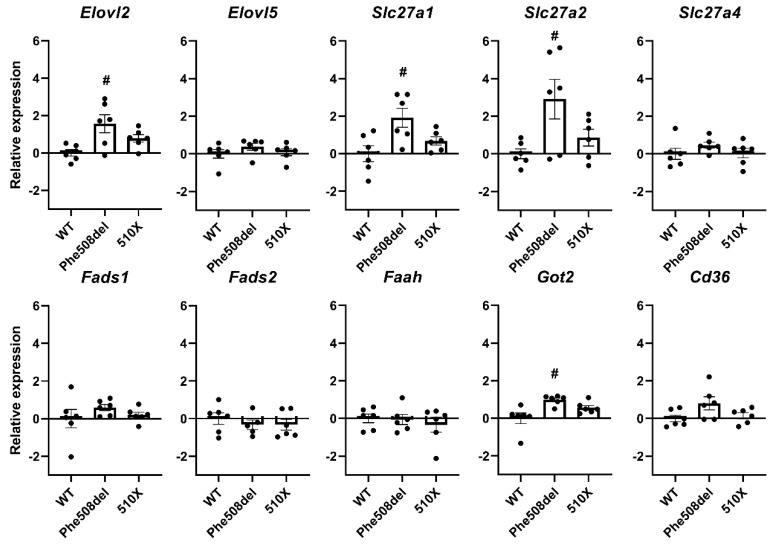
Expression of fatty acid synthesis and transport, and fibrotic mRNA in ileum of 1 month- ld CF rats. Expression of *Elovl2*, *Elovl5*, *Fads1*, *Fads2*, *Slc27a1, Slc27a2, Slc27a4, Faah, Got2, CD36*, *Col1a1* and *Col3a1* in wild-type, Phe508del and 510X rat ileum was normalized to the average expression of the housekeeping genes β-actin and cyclophilin A. The relative expression in the Phe508del and 510X ileum was compared to the wild type. Data are expressed as the mean ± SEM and analyzed by a one-way ANOVA with Dunnett’s multiple comparisons test to determine significance. # indicates *p* < 0.05, *n* = 5–6.

**Figure 3 ijms-24-07194-f003:**
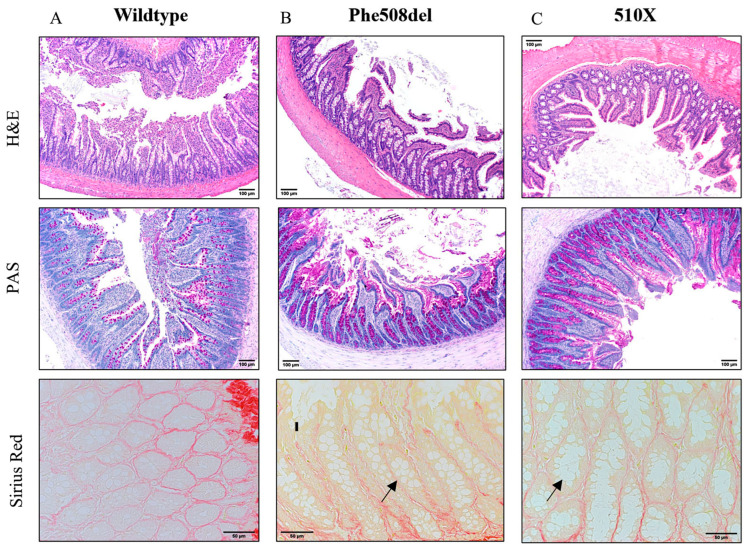
Histology in the ileum tissue of CF rat models. (**A**) Wild-type rats, (**B**) Phe508del rats, and (**C**) 510X rats. Scale bar is 100 μm (**A**,**B**) or 50 μm (**C**). *n* = 6. Arrows indicate mucus accumulation.

**Table 1 ijms-24-07194-t001:** Fatty acid composition in the serum of CF rat models.

	Wild Type		Phe508del		Knockout (510X)		*p* Values		
	2 Months	4 Months	2 Months	4 Months	2 Months	4 Months	Genotype	Age	Interaction
LA (18:2n-6)	22.9 ± 1.79	23.3 ± 1.67	27.5 ± 1.57	22.7 ± 1.57	24.1 ± 2.76	31.1 ± 6.97	ns	ns	ns
GLA (18:3n-6)	0.13 ± 0.01	0.11 ± 0.03	0.16 ± 0.03	0.14 ± 0.02	0.18 ± 0.06	0.26 ± 0.07	ns	ns	ns
AA (20:4n-6)	16.7 ± 2.25	15.8 ± 3.13	17.8 ± 1.56	18.0 ± 2.57	16.5 ± 2.18	15.7 ± 4.38	ns	ns	ns
DPA (22:5n-6)	0.10 ± 0.01	0.12 ± 0.01	0.09 ± 0.01	0.15 ± 0.04	0.10 ± 0.01	0.09 ± 0.02	ns	ns	ns
ALA (18:3n-3)	1.33 ± 0.24	1.47 ± 0.33	0.85 ± 0.11	1.36 ± 0.23	1.44 ± 0.18	0.96 ± 0.29	ns	ns	ns
EPA (20:5n-3)	0.94 ± 0.16	0.57 ± 0.20	0.95 ± 0.13	0.84 ± 0.13	1.26 ± 0.10	0.74 ± 0.17	ns	0.012	ns
DPA (22:5n-3)	0.67 ± 0.04	0.62 ± 0.15	0.35 ± 0.03	0.47 ± 0.07	0.63 ± 0.07	0.39 ± 0.04	0.031	ns	ns
DHA (22:6n-3)	3.25 ± 0.36	2.85 ± 0.32	2.34 ± 0.12	2.93 ± 0.41	3.16 ± 0.38	2.40 ± 0.34	Ns	ns	ns
LA:ALA	18.2 ± 1.99 ^a^	13.7 ± 1.84 ^b^	34.0 ± 5.15 ^c^	17.9 ± 2.58 ^a,b^	21.4 ± 1.30	23.2 ± 4.02	0.021	0.041	ns
AA:DHA	5.12 ± 0.17 ^a^	5.40 ± 0.51	7.75 ± 0.94 ^b^	6.18 ± 0.32	5.21 ± 0.16	6.29 ± 0.97	0.026	ns	ns
Palmitoleic acid (C16:1n7)	0.52 ± 0.10	0.75 ± 0.30	0.27 ± 0.02	0.41 ± 0.03	0.47 ± 0.10	0.32 ± 0.05	ns	ns	ns
Oleic acid (C18:1n9)	18.2 ± 2.10	19.4 ± 2.66	15.9 ± 0.32	17.7 ± 2.38	16.1 ± 0.96	15.4 ± 0.47	ns	ns	ns
Vaccenic acid (C18:1 trans 11)	2.21 ± 0.13	2.58 ± 0.49	1.69 ± 0.03	2.08 ± 0.20	1.75 ± 0.12	1.68 ± 0.31	ns	ns	ns
Elaidic acid (C18:1 trans 9)	0.09 ± 0.02	0.13 ± 0.02	0.09 ± 0.02	0.17 ± 0.04	0.15 ± 0.05	0.10 ± 0.01	ns	ns	ns
Eicosenoic acid (C20:1n9)	0.41 ± 0.05	0.43 ± 0.07	0.26 ± 0.02	0.37 ± 0.06	0.30 ± 0.06	0.30 ± 0.03	ns	ns	ns
Erucic acid (C22:1)	0.04 ± 0.01	0.03 ± 0.01	0.05 ± 0.01	0.04 ± 0.01	0.04 ± 0.01	0.03 ± 0.01	ns	ns	ns
Nervonic acid (C24:1n9)	0.18 ± 0.01	0.16 ± 0.02	0.17 ± 0.02	0.14 ± 0.02	0.21 ± 0.05	0.16 ± 0.05	ns	ns	Ns

LA: linoleic acid; GLA: gamma-linoleic acid; DGLA: dihomo-gamma-linolenic acid; AA: arachidonic acid; ALA: alpha-linoleic acid; EPA: eicosapentaenoic acid; DPA: docosapentaenoic acid; DHA: docosahexaenoic acid; LA:ALA: linoleic acid:alpha-linoleic acid ratio; AA:DHA: arachidonic acid:docosahexaenoic acid ratio. Data are expressed as the mean ± SEM and analyzed by a two-way ANOVA (genotype and age as factors) with Tukey’s post hoc test to determine significance. Superscript a is different to b is different to c, but not a,b. *n* = 6–8 per group. ns = not significant.

## Data Availability

The data presented here are available on request from the corresponding author.

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
