# Peer review of "Changes in Essential Fatty Acids and Ileal Genes Associated with Metabolizing Enzymes and Fatty Acid Transporters in Rodent Models of Cystic Fibrosis"

_ijms, 2023, doi:10.3390/ijms24087194_

Round 1

Reviewer 1 Report (New Reviewer)

In this study, Shrestha et al. studied the characterization of essential fatty acids (EFA), metabolizing enzymes, and fatty acid transporters in two rat models (Phe508del in CFTR and the other lacking functional CFTR (510X)) of cystic fibrosis. This study specifically focused on the ileum. The authors observed differences in fatty acid concentrations, metabolic enzymes, and fatty acid transporters in CF models with similar CF phenotypes. Due to the lack of a clinically mimicked model of CF about EFA status, the authors used novel models to see whether these two rats' models had similar EFAs status to CF patients. This is an interesting study with a novelty component. 

Here are my suggestions.

1) I am not quite sure why the authors looked at different parameters at different time points of study. Firstly, they looked into the fatty acid concentrations in 2nd and 4th-month-old rats (as mentioned in the methods section). However, all I see in table 1 is one set of data, and quite not sure whether this data is from 2nd-month-old rats or 4th-month-old rats. Secondly, they looked into the expression of metabolic enzymes and fatty acids transports in one-month-old rats and did not look at 2nd and 4th-month-old rats. Lastly, the histology was performed in 2nd and 4th-month-old rats (H and E staining and Sirius Red staining-for fibrosis) while the collagens expression was looked at one-month-old rats. Again, I am not quite sure whether the histology data is from 2nd-month-old rats or 4th-month-old rats. It is very difficult to interpret the results of the study. In order to interpret and conclude the results, all the results from all the points are crucial. 

2) Figure 3 and Table 2 data can be combined. Table 2 data can be presented as figures.

3) Look into the collagen's expression (Col1a1 and 3a1) at 2nd and 4th month old rats

Author Response

1) I am not quite sure why the authors looked at different parameters at different time points of study. ……. they looked into the expression of metabolic enzymes and fatty acids transports in one-month-old rats and did not look at 2nd and 4th-month-old rats.

The study was performed on biobanked samples. Unfortunately we did not have available tissue to do analysis on mRNA at 2 months and 4 months of age. This is why mRNA analysis is performed on 1 month old samples. We suggest that a future area of research should be the quantification of mRNA responsible for FA transport and metabolism across different ages (Line 317+).

…… they looked into the fatty acid concentrations in 2nd and 4th-month-old rats (as mentioned in the methods section). However, all I see in table 1 is one set of data, and quite not sure whether this data is from 2nd-month-old rats or 4th-month-old rats.

This has been amended to separate the 2 month old data from the 4 month old data.

Lastly, the histology was performed in 2nd and 4th-month-old rats (H and E staining and Sirius Red staining-for fibrosis) while the collagens expression was looked at one-month-old rats. Again, I am not quite sure whether the histology data is from 2nd-month-old rats or 4th-month-old rats. It is very difficult to interpret the results of the study. In order to interpret and conclude the results, all the results from all the points are crucial. 

We have reanalysed the data and updated the data to acknowledge that only 2 month old histology data is presented.

2) Figure 3 and Table 2 data can be combined. Table 2 data can be presented as figures.

To make the manuscript more clearer, we have added Col1a1 and Col3a1 mRNA expression to Figure 2. We have removed Table 2, as this data is the text (Line 213-215).

3) Look into the collagen's expression (Col1a1 and 3a1) at 2nd and 4th month old rats

The study was performed on biobanked samples. Unfortunately we did not have available tissue to do analysis on mRNA at 2 months and 4 months of age. We suggest that a future area of research should be the quantification of mRNA responsible for FA transport and metabolism across different ages (Line 317+).

Reviewer 2 Report (New Reviewer)

This study examines the fatty acid profile in the ileum from two CF rat models (F508Del and KO) compared to WT controls. Even though similar histopathology changes were observed in the ileum of these two rat models, aberrations in certain fatty acid species differed. In addition, Cftr mRNA abundance and the abundance of mRNAs that encode factors involved in fatty acid metabolism also differed between the two CF rat models. This is the first characterization of essential fatty acids in CF rats.

While the methodology of this study is sound and well-described, it is difficult for a reader to understand the reason behind some of the analyses because no context is provided. There is also a lack of clarity in the description of the rat models and some of the endpoints examined. This lack of information needs to be provided.  I offer more details below:

1. The KO rat model needs further description.  For example, it is not clear that the 510X model is actually caused by an 8 nucleotide deletion in exon 11 that results in the formation of a premature termination codon and not from  a nonsense mutation.

2. In the abstract, it is not clear that examination of Cftr and the fatty acid metabolic factors are at the mRNA level. 

3. In Figure 2, it is unclear what role each of the genes examined plays in fatty acid metabolism.  A brief description or even a table that shows exactly how these genes might affect fatty acids would be helpful.

4. Similarly, it is unclear why Col1a1 and Col3a1 are examined in Figure 3. Some context about why you are specifically examining the abundance of these mRNAs would be beneficial to understanding your data results and interpretation.

5. A brief review of what has previously been found concerning fatty acids in other CF animal models would be helpful. While some information is provided in the discussion, it seems choppy and it is difficult to get a full picture of differences that occur in fatty acids among different CF animal models, which is important since they vary even between rat CF models. 

6. A more thorough explanation of the differences found between the two CF rat models is warranted. For example, while Cftr mRNA abundance is discussed, there is no discussion of potential differences CFTR function between the two rat models? Is there truncated 510X protein expressed?

7. The authors suggest that the fatty acid profiles in the CF rat differ from what is observed in CF patients.  Can they offer the these differences might occur? Does this mean that the CF rat is not a good model to examine fatty acids in the context of CF because of these differences?

8. The authors don't examine fatty acid profiles in the lung, even though they do briefly mention difference in the expression of factors associated with fatty acid metabolism in human bronchial epithelial cells. Why did the authors not also examine the fatty acid profile in the rat lungs? Why was the examination limited to the ileum in rats?

9. Could the authors explain how altered fatty acid metabolism is related to CF disease manifestations in different tissues? In the introduction, they site a study suggesting that EFA deficiency is independent of pancreatic insufficiency. So how does EFA deficiency fit into the various CF manifestations? 

Author Response

While the methodology of this study is sound and well-described, it is difficult for a reader to understand the reason behind some of the analyses because no context is provided. There is also a lack of clarity in the description of the rat models and some of the endpoints examined. This lack of information needs to be provided.  I offer more details below:

We appreciate the comments and have added the responses below.

1. The KO rat model needs further description.  For example, it is not clear that the 510X model is actually caused by an 8 nucleotide deletion in exon 11 that results in the formation of a premature termination codon and not from  a nonsense mutation.

We appreciate the comment and have added this information into the new version of the manuscript (lines 100-104)

2. In the abstract, it is not clear that examination of Cftr and the fatty acid metabolic factors are at the mRNA level. 

Again, we appreciate the comment and have added this information into the new version of the manuscript (line 28, 29)

3. In Figure 2, it is unclear what role each of the genes examined plays in fatty acid metabolism.  A brief description or even a table that shows exactly how these genes might affect fatty acids would be helpful.

A description of some of the targets eg FADS1, FADS2, ELOVL2 and ELOVL5 is described in Lines 70 - 74. We have added additional description of the other targets in lines 60-63.

4. Similarly, it is unclear why Col1a1 and Col3a1 are examined in Figure 3. Some context about why you are specifically examining the abundance of these mRNAs would be beneficial to understanding your data results and interpretation.

This has been added to lines 214-216.

5. A brief review of what has previously been found concerning fatty acids in other CF animal models would be helpful. While some information is provided in the discussion, it seems choppy and it is difficult to get a full picture of differences that occur in fatty acids among different CF animal models, which is important since they vary even between rat CF models. 

We appreciate the comment and have published a recent review [10]. We have added more information in the introduction (lines 79-90) and discussion (lines 260-263) about this.

6. A more thorough explanation of the differences found between the two CF rat models is warranted. For example, while Cftr mRNA abundance is discussed, there is no discussion of potential differences CFTR function between the two rat models? Is there truncated 510X protein expressed?

We have added to the discussion of the phenotype for the rats is in lines 101-105 as cited from our previous publication [15]. We have added additional information about the protein expression as cited from our previous publication [15] (Lines 89-90).

Round 2

Reviewer 1 Report (New Reviewer)

N/A

This manuscript is a resubmission of an earlier submission. The following is a list of the peer review reports and author responses from that submission.

Round 1

Reviewer 1 Report

The study by Shrestha et al. demonstrate changes in fatty acids in a CF mutant rat line. This is significant in that it establishes a model for characterizing the relationship between CFTR expression and fatty acid transport and metabolism. Overall the manuscript is well written, but could benefit from clarifying a few concepts. For example, neither the abstract nor introduction state the function of cftr, only the disease phenotype. An upfront statement of protein function might be helpful to the reader. A summary or introduction figure that graphically captures the relationship between wt, cftrdel and cftrx would also be helpful to summarize findings.

Author Response

  • Thus it is a pity they summarize results from 1-4 months old animals and the dietary intake is only very briefly described, hazarding the interpretation of the results.

Unfortunately, due to limited availability of samples, it was necessary to combine data from animals at different ages. To make interpretation of the results clearer, we now have included more specific information in the manuscript methods stating which analyses were performed on which age animals.

To provide more information about the dietary intake we have now included a supplementary table listing the proportion of fatty acids included in the rodent diet.

  • The discussion in very wordy.

The discussion and expression have been improved.

  • Title; The title indicates conclusive results which are not obtained.

We appreciate the comment and acknowledge that for the rodent model genotypes have differences in essential fatty acid transporters and enzymes, but not the fatty acids themselves. We have amended the title.

  • Introduction - line 40. Today´s view is that some of the CFTR protein probably reach the membrane but most are lost as described.

This has been amended

  • Material and Methods; - line 85. A table with the dietary fatty acids should be included; also should be explained if all rats had been on the same diet since weaning?? Or birth (via mothers?)

We have updated this section to state that “From birth, all animals were maintained on a 50:50 mix of normal (6.5%) and high-fat (9%) irradiated rodent chow”.

We have now included a supplementary table listing the proportion of fatty acids included in the diet.

  • Line 90 - what is ¨humanely euthanised¨??

Our language was not clear here and we have now changed it to humanely killed (i.e. an animal may be defined as one in which the animal is rendered unconscious, and thus insensitive to pain, as rapidly as possible with a minimum of fear and anxiety.)

  • Line 93 - If serum fatty acids were analysed must be given time of fasting since not using RBC or phospholipids indicate that the results are heavily dependent on the food intake before sampling which can be a major factor for the non-conclusive results of the fatty acid analyses.

The animals were not fasted as the serum samples were taken from a biobank that was not established specially for nutrition studies. In our previous studies using a different rodent model, we have published non-fasted fatty acids.

https://doi.org/10.1113/JP277583

We have noted in the methods fatty acid analysis section that animals were non-fasted and we have also added a point in the discussion section to address this limitation: “Notably, the rats in this study were non-fasted, therefore the fatty acid analyses may have been influenced by food consumption prior to blood sample collection”.

The rats were all killed (and serum samples collected) in the light cycle, which is when they graze less. The removal of food during the light cycle may have caused some distress which would have affected results and having the rats fasted would have also impacted other results.

Further, the analysis is compared to wildtype rats under the same conditions ie non-fasted, collection in light cycle.

  • Line 123 – It is unexpected that collagen staining is used for ileum. A motivation would be valuable.

This has now been added.

  • Line 127 – Kruskall-Wallis would be more appropriate with this small material. Median and interquartile range would similarly be more appropriate than mean since some of the SEM are very high.

We performed Shapiro-Wilk tests on all datasets and this indicated that the data were normally distributed. Given this, it is only appropriate to perform parametric post-hoc tests, such as the Tukey’s or Dunnett’s.

We have opted to keep the data expressed as mean and SEM as this format is easily interpretable to the reader as it is the most common way of reporting data in the field. While the SEMs are (somewhat) high for some of the datasets, we do not believe that this is a valid reason to avoid using them.    

  • Results;- line 134 - DPA shall be defined (n-3) because the reader not familiar with fatty acids might refer it to n-6( correct in the table). The monounsaturated fatty acids and Mead acid shall be shown as well, because they are important in the CF lipid pattern.

These have been included. Unfortunately Mead acid was not investigated.

  • Line 136 -  How would the authors explain the increase of LA as well as the increase of LA/ALA if not the diets are shown?  I am quite aware it is not significant but the trend is very clear for the 510X group ((very high SEM). With this very low number it would be more appropriate to use median (see above). Are the distribution of values related to the age of the animals?? And are the ages similar in all groups? Sex differences??

We have now included the diets as part of the manuscript (supplementary S1). Animal ages in all groups was similar. The question of sex is interesting, and should form the basis of future studies. The concentration of LA is 3.25%, which is similar to a standard chow diet. Please see comment above regarding SEM.

  • Figure 2 is duplicated

The second Figure 2 has been deleted

  • Line 172. How to explain no increase of PAS staining but mucus extrusion into the lumen in Figure 3? Were the analyses performed in rats of different ages? Also mentioned in the discussion line 198

The histological analyses were performed on a cohort of rats that were 2 and 4 months of age. Due to limited animal availability, the samples from these two time points were combined and classified as “adult rats” as they are >2 months of age.  

The image quantification revealed that there was no significant difference in PAS staining between the genotypes. We suspect that while the localisation of mucus differs between the genotypes, the overall amount of mucosubstances are similar. The CF rats demonstrated high levels of mucus extruded into the intestinal lumen, while wild-type rats only occasionally exhibited luminal mucus. The images shown in Figure 3 demonstrate only one example of PAS staining from each genotype and do not capture the heterogeneity in mucus levels between animals.

We have also revised line 198 to state that “In the 510X rats the increase in villi height and mucus accumulation in the intestinal lumen is also indicative of the pathology observed in CF patients”, thus we have focussed only on the fact that there was increased mucus accumulation in the lumen of the intestine, not an overall increase in mucins. 

  • Figure 3 is not only showing histology. Also the RNA expression is shown in D and E (missing in the legend)
  • Table 2 is indicated to give results from liver, which is missing (also in the text).

Liver has been removed from the table text

  • Discussion; - line 192 How can the Elvol2 expression increase explain the increased ratio?? Neither of the long-chain n-3 fatty acids are increased??

This has been discussed further in the discussion

  • Line 204 – Also the ferrets had low LA in ref 19, but not in RBC.

This has been amended

  • Line 232- The authors have not investigated ELOVL6 and the mention of this here is just out of the context, especially when they have not shown any saturated of monounsaturated fatty acids.

This has been removed

  • Line 235 – What is the reason to postulate that Elovl 2 is preferentially metabolizing DPA and DHA over AA?? There might be other explanations!

This has been suggested now, and not a definitive link

  • The paragraph about sex dimorphism is, lines 259-269, irrelevant since any sex differences are not shown.

This has been removed

  • The conclusion should be much humbler. Their data are not convincing but interesting.

This has been amended

Reviewer 2 Report

 This paper reports fatty acid concentrations and ileal histology and enzyme RNA expression in rat models of CF, the Phe508del and a class one model (510X). It is an interesting new model which don´t present many of the typical CF symptoms but might therefore be suitable for long-term studies to follow the disease development. Thus it is a pity they summarize results from 1-4 months old animals and the dietary intake is only very briefly described, hazarding the interpretation of the results. The discussion in very wordy.

Title;  The title indicates conclusive results which are not obtained.

Introduction - line 40. Today´s  view is that some of the CFTR protein probably reach the membrane but most are lost as described.

Material and Methods; - line 85. A table with the dietary fatty acids should be included; also should be explained if all rats had been on the same diet since weaning?? Or birth (via mothers?)

Line 90 - what is ¨humanely euthanised¨??

Line 93 - If serum fatty acids were analysed must be given time of fasting since not using RBC or phospholipids indicate that the results are heavily dependent on the food intake before sampling which can be a major factor for the non-conclusive results of the fatty acid analyses.

Line 123 – It is unexpected that collagen staining is used for ileum. A motivation would be valuable.

Line 127 – Kruskall-Wallis would be more appropriate with this small material. Median and interquartile range would similarly be more appropriate than mean since some of the SEM are very high.

Results;- line 134 - DPA shall be defined (n-3) because the reader not familiar with fatty acids might refer it to n-6( correct in the table). The monounsaturated fatty acids and Mead acid shall be shown as well, because they are important in the CF lipid pattern.

Line 136 -  How would the authors explain the increase of LA as well as the increase of LA/ALA if not the diets are shown?  I am quite aware it is not significant but the trend is very clear for the 510X group ((very high SEM). With this very low number it would be more appropriate to use median (see above). Are the distribution of values related to the age of the animals?? And are the ages similar in all groups? Sex differences??

Figure 2 is duplicated

Line 172. How to explain no increase of PAS staining but mucus extrusion into the lumen in Figure 3? Were the analyses performed in rats of different ages? Also mentioned in the discussion line 198

Figure 3 is not only showing histology. Also the RNA expression is shown in D and E (missing in the legend)

Table 2 is indicated to give results from liver, which is missing (also in the text).

Discussion; - line 192 How can the Elvol2 expression increase explain the increased ratio?? Neither of the long-chain n-3 fatty acids are increased??

Line 204 - Also the ferrets had low LA in ref 19, but not in RBC.

Line 232- The authors have not investigated ELOVL6 and the mention of this here is just out of the context, especially when they have not shown any saturated of monounsaturated fatty acids.

Line 235 – What is the reason to postulate that Elovl 2 is preferentially metabolizing DPA and DHA over AA?? There might be other explanations!

The paragraph about sex dimorphism is, lines 259-269, irrelevant since any sex differences are not shown.

The conclusion should be much humbler. Their data are not convincing but interesting.

Author Response

  1. In figure 1, Cftrgene expression in wildtype show a huge range. Did it only happen in the ileum? What’s the reason? Is Cftr gene expression similar to other tissues?

The large variation in cftr mRNA expression was also observed in wild type lung tissue (McCarron et al 2019). Large variations are normal for in vivo data due to animal-animal variability. Only a small section of the ileum was used for RNA isolation and therefore the level of expression could be different depending on what part was used.

We have added some more detail regarding cftr mRNA expression in cells/tissue based on a recent 2023 publication.

  1. In line 169-177, it should not be bold;

This has been reformatted

  1. In line 176, add ‘,’ after ‘Col3a1’;

This has been amended

  1. In figure 3, the labels and arrow of panel A, B, and C are not right;

This has been amended

  1. In line 181, the information of scale bar is not right;
  2. In line 203, delete dot ‘.’ after ‘reduction’;

This has now been removed

  1. In line 205, add dot ‘.’ after ‘[19]’;

This has now been amended

  1. In line 222, ‘ileumhowever’ should be ‘ileum. However’;

This has now been amended

  1. In line 223, delete one dot ‘.’ after ‘phenomenon’;

This has been amended

  1. In line 238, add ‘,’ after ‘however’;

This has now been amended

Reviewer 3 Report

The focus of this paper is trying to characterize the changes of essential fatty acid (EFA) in cystic fibrosis (CF) rat models caused by CFTR mutant Phe508del or 510X. The decrease of DPA and increase of AA:DHA in CF rat model (Phe508del) are similar to CF patients. The expression increase of genes responsible for EFA metabolic enzymes and transporters contributed to the serum concentrations of EFA. This study also revealed the relationship of phenotype changes in CF rat models and EFA deficiency. In short, this study reported a CF rat model with EFA deficiency for the first time.

1.      In figure 1, Cftr gene expression in wildtype show a huge range. Did it only happen in the ileum? What’s the reason? Is Cftr gene expression similar to other tissues?

2.      In line 169-177, it should not be bold;

3.      In line 176, add ‘,’ after ‘Col3a1’;

4.      In figure 3, the labels and arrow of panel A, B, and C are not right;

5.      In line 181, the information of scale bar is not right;

6.      In line 203, delete dot ‘.’ after ‘reduction’;

7.      In line 205, add dot ‘.’ after ‘[19]’;

8.      In line 222, ‘ileumhowever’ should be ‘ileum. However’;

9.      In line 223, delete one dot ‘.’ after ‘phenomenon’;

10.  In line 238, add ‘,’ after ‘however’;

Author Response

  1. It is necessary to do analysis for the protein expression of the genes responsible for EFA transport and metabolism and CFTR in intestinal tissues.

The authors agree that this would provide further data that would help understand EFA transport and metabolism but would be better suited to the next manuscript further looking at these pathways. We have previously looked at IHC CFTR expression in the lungs of these rats (McCarron et al., 2019). We have made mention of further studies in the future should look at protein expression in the discussion.

  1. CF Phe508del and 510X rats have the same phenotypes, but have different EFA change. It needs to explain in the discussion. Could you please do further experimental study to explore the cause of the difference?

There are differences between the Phe508del and 510X rat phenotypes which have been published (McCarron et al, 2019). We are currently in the process of unravelling other pathways that are differentially affected in the two CF rat models, however believe that the inclusion of further data regarding this in the manuscript would dilute the focus and aims of this manuscript and thus decrease the clarity and impact on the reader.

An explanation for the differences between Phe508del and 510X rats can be found on lines 219-223.

Reviewer 4 Report

Patients with CF routinely have deficiencies in essential fatty acid (EFA).  The study for the molecular mechanism that gives rise to EFA deficiency in animal models of CF were limited. The study analyzed the alteration of circulating EFA used a newly-developed CF rat that mimic what is observed in patients with CF and found the changes in EFA concentrations due to altered expression in genes responsible for EFA transport and metabolism, and histological changes to the ileum, similar to patients with CF. The study contribute to explains the EFA deficiency in CF patients.

Question

1. It is necessary to do analysis for the protein expression of the genes responsible for EFA transport and metabolism and CFTR in intestinal tissues.

2. CF Phe508del and 510X rats have the same phenotypes, but have different EFA change. It needs to explain in the discussion. Could you please do further experimental study to explore the cause of the difference?

Author Response

For example, neither the abstract nor introduction state the function of cftr, only the disease phenotype. An upfront statement of protein function might be helpful to the reader. A summary or introduction figure that graphically captures the relationship between wt, cftrdel and cftrx would also be helpful to summarize findings.

We have now added a brief description in the introduction concerning the function of CFTR.

In the conclusion we have clarified the differences between wildtype and the 2 mutants. 

Round 2

Reviewer 2 Report

The authors have improved the manuscript but still there are important lack of information. In the manuscript is described the ages of the animal investigated (improvement!) but not the number and not if there were differences in fatty acids or histology at different ages. Were the presented results a mixture of different ages?? If so – were there no differences which justified mixed analyses or was it only too few cases?  Furthermore, the trans fatty acid are not correctly identified and why are they not reported as fatty acids?? Do they were analysed with different methodology?? The paragraph about the action of Elovl2 (in many cases wrongly spelled) is too speculative.

Minor:

Title is misleading. The paper is not conclusive in either fatty acids or enzymes.

Line 143: p>0.05 is not significant.

Lines 251ff wrong spelling of ELOVL2

Table 1. What age of the animals? Number? Why are not all fatty acids? Trans should be indicated.

Figure 2. The significance marker in text and figure are not congruent.

Discussion: Line 200. How can the increase of expression of ELOVL2 in ileum explain a ratio in serum? It is to draw to much conclusion from a single observation. Ileum is the most affected intestinal part in CF, thus an increased expression of CFTR and fatty acid transporting enzymes seems very unlikely and thus need to be scrutinized or explained. Furthermore, the high n-6/n-3 ratio in the diet might be a factor explaining the metabolism with high AA/DHA ratio.

Reference 19 is missing an author.

Supplementary tables have mixed up the titles.

Author Response

Please see the comments addressed in the attached file. This is for the reviewer and editor

Reviewer 4 Report

the manuscript has been sufficiently improved

Author Response

(The authors gave the same response as above.)

Round 3

Reviewer 2 Report

I cannot see that the authors have responded to my questions and remarks.

Studying the revised manuscript shows that they have only partly changed according to recommendations. I repeat and complete my suggestions:

Title.

Not appropriate since no deficiency or abnormality in essential fatty acids are shown by the authors. With the small number of animals of different ages, a so weak significance for one ratio can not be taken as any proof for characterization of fatty acid profile of the rats.

In abstract is written docospentaenoic acid without telling which kind of omega fatty acid they refer to. It is the only significantly different fatty acid but is never commented in the paper.

Introduction. 

Is written that CFTR is rapidly degraded in the membrane, I have not seen any proof for that.

Lines 49-50. Triglycerides undergo digestion, not EFA.

Line 52 adipose what??

Line 63. ALA is not more abundant in tissue, but in food, but that is not relevant in this context.

Methods.

The method for fatty acid analyses is not properly described. Serum cannot be analyzed on GC. There are several extraction steps before you get FAME which you can analyze on GC, which together with any reference to method should be explained.

The fatty acid composition of the diet is not given but is important because that might explain the high amount of trans-fatty acids the authors have found.

Table 1.

Names of many fatty acids in the Table 1 has not been changed from salts to fatty acids, only elaidic acid has been properly changed. You cannot analyze salt on GC.

Furthermore, to make a mean of the values from different ages is not appropriate without showing that no differences are found. The very low number of individuals make it difficult but the mean is not interesting. Additionally, palmitoleic and oleic acids show contrary pattern to that shown in CF  with low LA, both in humans and animals, indicating that the rat model has not deficiency of LA which is usually seen in other models (not all mice models). No Mead acid was detected, or?? The authors try to make too much conclusions of two weak data.

Author Response

Title.

Not appropriate since no deficiency or abnormality in essential fatty acids are shown by the authors. With the small number of animals of different ages, a so weak significance for one ratio can not be taken as any proof for characterization of fatty acid profile of the rats.

This has been amended

In abstract is written docospentaenoic acid without telling which kind of omega fatty acid they refer to. It is the only significantly different fatty acid but is never commented in the paper.

This has been amended to include the updated data.

Introduction. 

Is written that CFTR is rapidly degraded in the membrane, I have not seen any proof for that.

A new reference [3] is added.

Lines 49-50. Triglycerides undergo digestion, not EFA.

This has been amended lines 48-49

Line 52 adipose what??

Tissue line 51

Line 63. ALA is not more abundant in tissue, but in food, but that is not relevant in this context.

Line 67 indicated that metabolism occurs via enzymes in tissues, and it is not clear where this statement comes from.

Methods.

The method for fatty acid analyses is not properly described. Serum cannot be analyzed on GC. There are several extraction steps before you get FAME which you can analyze on GC, which together with any reference to method should be explained.

This has been updated.

The fatty acid composition of the diet is not given but is important because that might explain the high amount of trans-fatty acids the authors have found.

In the supplementary file submitted with the previous version had the fatty acid composition. No trans fats were in the diet. The concentration of trans fatty acids across the group did not differ.

Table 1.

Names of many fatty acids in the Table 1 has not been changed from salts to fatty acids, only elaidic acid has been properly changed. You cannot analyze salt on GC.

This has been amended Table 1.

Furthermore, to make a mean of the values from different ages is not appropriate without showing that no differences are found. The very low number of individuals make it difficult but the mean is not interesting.

We have separated the data based on age for fatty acids and performed a 2 way anova with genotype and age as the 2 variables.

Additionally, palmitoleic and oleic acids show contrary pattern to that shown in CF with low LA, both in humans and animals, indicating that the rat model has not deficiency of LA which is usually seen in other models (not all mice models).

No changes in palmitoleic acid and oleic acid, and no changes in LA. We have reanalysed the data based on 2 way analysis of genotype and age. We agree that the rat model, at the ages studied, do not have LA deficiency. There is an increase in AA/DHA in Phe508del at 2 and 4 months, which is similar to humans with CF.

No Mead acid was detected, or??

Mead acid was not measured using the technique provided by Cardinal Biosciences